# Anthropocentrism, Natural Harmony, Sentience and Animal Rights: Are We Allowed to Use Animals for Our Own Purposes?

**DOI:** 10.3390/ani13061083

**Published:** 2023-03-17

**Authors:** Giuseppe Pulina

**Affiliations:** Agraria Department, University of Sassari, 07100 Sassari, Italy; gpulina@uniss.it

**Keywords:** animal sentience, animal rights, anthropocentrism

## Abstract

**Simple Summary:**

The attention of scientists and citizens is increasingly focused on the issue of animals as sentient beings. This inherent quality of animals, although recognized by many countries at a regulatory level, if not well defined in scientific terms can lead to misunderstandings and the wrong consequences. It is important to note that animal sentience, understood as the ability to experience emotions, is limited in extent and restricted to certain animal species. While this understanding may lead some to believe in universal animal rights, this is not logical as the recognition of animal life free from fear, restriction, and suffering is actually a human right, not an animal one.

**Abstract:**

Taking a cue from J.W. Yates’ recent work on animal sentience published in this journal, which explores the field and categorizes it as a harmony with nature and a recognition of its values, inferring that the inclusion of animals in the sphere of objective rights is the obligatory step for a real sustainability in all human activities, this opinion paper seeks to challenge some of the claims made in the article and present an alternative perspective on sentience and animal rights. Preliminarily, I propose a semantic word-washing and the use of more precise terms instead of not well-defined ones such as “harmony” in relation to nature and “sentience” in relation to animals, and I affirm that there can be only one point of view, however rich in dialectics, which is the human one for looking at the problems of animal ethics. Below, I present the thesis that concludes that it is not possible to attribute rights to animals, but it is our right and duty to protect their well-being, which requires states to pass laws for their defence. I conclude that while it is acceptable to raise animals for priority human rights (such as food and health), it is also an obligation to properly care for and protect them.

## 1. Introduction

In this paper, I aim to challenge some of the claims made in the article recently published in this journal [1], and offer a differing perspective on animal sentience and rights. Although the article appears to serve as a manifesto for the World Federation of Animals, its publication in a scientific journal highlights the importance of developing the debate by discussing these issues in animal ethics and consequent morality. However, as is well known, scientific knowledge derives from the application of the laws of hypothetical-deductive thinking, which can be summarised as (a) formulation of a hypothesis, (b) deduction of certain phenomena as consequences from this, and (c) experimental verification of these consequences, which if positive, do not reject the hypothesis, but neither confirm it, as mistakenly believed by many, until proven otherwise [2]. Given the opinion-paper nature of this work, I will instead *predominantly* use the philosophical method and rely on the sound principle of non-contradiction [3], while attempting to delineate a hypothetical-deductive reasoning using a language as free of semantic ambiguities as possible.

## 2. Epistemic Premises: Semantic Cleaning

The term anthropocentrism has become increasingly used to describe the prevalence of humans over other natural systems. This term has been used with regard to the human-animal relationship and its harmful consequences not only for animals, but also for the natural environment and climatic impact (among many, [4]). Some authors go so far as to consider anthropocentrism, declined in the form of urbicide, as a condition of unlimited human violence towards the environment and other human beings [5].

The philosophical debate is also heated on the anthropomorphic and anthropocentric fallacy with respect to the application of these designations in science and environmental matters [6] but goes beyond the scope of this brief paper.

In short, although the debate on whether nature should be human-centred or not remains open, there is no doubt that, among beings, humans are the only with the ability to express a point of view, due to the possession of a language capable of communicating abstract concepts. This means that other species endowed with their own, sometimes sophisticated, language can communicate with each other, even at considerable distances, but until proven by experimental evidence, they are not capable of generating complex linguistic structures such as human languages can produce.

As such, the only possible perspective on the role of humans in nature is the human perspective. In this sense, it is misleading to speak, usually in negative terms, of anthropocentric viewpoints, because there are no alternatives. If, on the other hand, the discussion shifts to determining who should be at the centre of reflection and action, there is a risk of falling into extreme relativism, where everything in the universe can be considered either central or peripheral, or ecumenically both.

Regarding anthropomorphism in the natural sciences, we only need to recall Sullivan’s [7] warning: “Because of our natural tendency to interpret the world in the light of human experience, and because language expressing this viewpoint comes most readily to our tongues, it is a constant intellectual discipline when talking about animal behaviour, to steer clear of anthropomorphism”. In short, when we speak of ’animal feelings’ we can only use terms that refer to man, but this poses a semantic problem for the objectivity of scientific observations. However, the closer the animals get to the evolutionary lines of the hominins, the greater similarities can be traced in the behaviours that are mirrors of the mental representations possessed by many cognitively complex animals. Consequently, the definition of sentience as the ability of animals to experience emotions is influenced by anthropomorphism and, if reported in scientific papers, is not very precise, unless it is admitted that it is provisional and that it will be replaced by more correct terms based on progress of cognitive science.

The third argument regards the presence of values in nature, as asserted by Yates [1], who claims that human actions towards nature cause harm and suffering to animals, which represents a negative value for them. However, it is important to note that the term ‘nature’ comes from the Latin word *natūra*, the future form of the verb *nasci*, ‘to be born’ and it originally referred to the Romans’ understanding of what the Greeks referred to as “physis,” or reality, as distinct from meta-physis. To assign values to nature *per se* is therefore a profoundly flawed concept, as nature does not have a moral code, and therefore, there is no ethical meaning in its actions. The distinction between right and wrong is a human construct and only applies to human affairs. Inflicting pain on prey is not considered right or wrong, it is just a part of nature. On the other hand, causing suffering to domestic or pet animals is considered wrong, not because of an inherent negative value in nature, but because it goes against human values. However, from a strictly utilitarian point of view, it cannot be denied that natural resources represent a primary value for animals and, probably, in this sense [1] ascribes this value to them and, by transition of meaning, elevates it to universal value recognized by humans.

Furthermore, the use of terms such as ‘harmony’ or ‘balance’ in reference to natural facts is wrong when applied in scientific contexts: the former is borrowed from the language of music and indicates a subjective judgement about an object under evaluation, while the latter simply does not exist since ecosystems are subject to continuous transformations and are never in balance [8].

Lastly, the greatest confusion that arises when talking about animals is the type of animal considered. It is usually taken for granted that one is talking about animals with a complex nervous system, assuming by analogy that they can ‘feel the world’ as we perceive and interpret it. However, this implies that we must identify a threshold below which the category of sentience is no longer valid. Unfortunately, the threshold for attributing mental abilities to animals has not been clearly fixed. As more knowledge is gained, scientific communities become increasingly cautious about making such attributions to animals (from limbal to primates), relying on a scale of complexity to determine which animals fall above or below this threshold. This makes it impossible to espouse the position of some scholars, including Mikhalevich and Powell [9], that all animals, vertebrates and invertebrates, should be included in the category of beings to be protected, a kind of absurd pan-animalism whereby parasites and disease vectors should also be protected.

## 3. Formulation of a Hypothesis

Having argued that the only perspective that can be discussed here is the anthropocentric one, understood as the human point of view, rejecting the notion of harmony and redefining that of values in nature, and highlighting the issues with using the term “sentience” in a scientific context, I aim to demonstrate that using rigorous language and the scientific method supports the thesis that it is acceptable to raise and slaughter animals (in this case specifically referring to those species used for food production, but the reasoning can also apply to pets, lab animals, and sports animals with necessary adjustments), and that animals do not possess moral status and thus, do not have subjective rights. As previously mentioned in my works, I unequivocally declare the obligation of humans to reduce animal suffering to zero, when possible, and ensure their complete welfare while under human care one for the sake not only of humans but of animals themselves [10,11].

## 4. Is It Permissible to Raise and Sacrifice Animals for Our Own Needs? Some Premises

Anecdotally, most people provide one of two diametrically opposed answers, namely: “yes” or “no”, while a small minority, after careful reflection, responds: “maybe”. Many works have explored the public’s propensity towards the application of animals for pharmaceutical and medical experimentation [12] and an immense literature has been devoted to the welfare of domestic [13], or wild animals in captivity [14] or free [15].

The proponents of the “yes” will try to justify their position with arguments that refer to the prevalence of the human species over others [16] (basic in many religions, but not in all), to utilitarianism (preferred by those who have a secular approach) [17], to tradition [18], to the evolution of our species [19] and, finally, to the reciprocity of advantages for human-animal contractors [20]. The supporters of the “no” (a more comfortable side of the contemporary dialectic) will restrict the arguments to anti-speciesism (basic in many disciplines of holistic thought) and to the denial of any utility in the interspecific human-animal interaction [21], with the aggravating circumstance of the sufferings and killing of the latter to satisfy the needs of the former [22] (basic in the animalist ideology-religion [23], with serious collateral reverberations in the more recently coined ecologist one). Finally, the small minority of “perhaps” will begin to put together a chain of “distinctions” which can hardly be included in this short paper, but which are the fruitful sign of a debate that is still open and which will unlikely find an ultimate answer. In fact, if following the teachings of Floridi [24], philosophy is rather the science of questions asked well than of the relative answers, albeit adequate to the quality of the questions. The themes of ethics and morals applied to our relationship with animals pose many questions which are, and will be, difficult to answer.

The ethics of human-animal relations is considered one of the most controversial and ancient subjects of human thought, starting with the Pythagoreans [25] and ending with contemporary moral philosophers [26], and anticipates, by several centuries, the one now emerging which concerns artificial intelligence as an alien entity capable of self-determination even to the detriment of our species [27], although not all scholars agree on the possibility that AI can override the qualities of human feelings [28].

Returning to our question of whether it is permissible to raise and sacrifice animals to our own ends, I restricted the field of this opinion paper to those animals bred for the production of food, leaving out other cases, although logically connected, such as circus animals, competition, pets, for military or scientific use, for zoological gardens, present in natural environments infested with tourists, for decoration (see goldfish, among others).

The fact of being an animal scientist places me among those who initially answer the question with a “maybe” and then move resolutely towards answering “yes”. This stance is necessary to effectively argue the reasons why I, and others, believe that breeding animals is acceptable. This answer cannot be based purely on philosophical principles, i.e., open to the principle of non-contradiction with Floridi’s [24] position mentioned above, but must be based on operational and applied ethics to determine what is right or wrong and this is the role of applied ethics. Similarly, it is the role of morality to determine whether societies that engage in animal breeding are reprehensible.

A particular aspect not to be underestimated when talking about domestic animals, even if not for food production, is that these are human artefacts in the sense that they have undergone a millennia-long selection of temperament, so most of them are far from behaving like their natural ancestors [29,30], at least as long as they remain under human control, to revert, sometimes rapidly, to wild behaviour once they are left in the wild.

I suspend the discussion on this point to introduce an argument that is essential to support what has been said, namely that breeding animals for our own purposes is morally acceptable.

## 5. Do Animals Have Rights?

To answer, I have assumed the concept of right elaborated by the Stanford Encyclopedia of Philosophy in 2020: “*Rights are entitlements (not) to perform certain actions, or (not) to be in certain states; or entitlements that others (not) perform certain actions or (not) be in certain states*”. However, to ascertain whether animals are subjects of rights or bearers of rights, we must make a premise regarding the possibility of animals themselves taking an active part in the social contract.

I will therefore analyse the concepts of “perception”, “consciousness”, “awareness”, and “ability to elaborate mental representations”. All living organisms are perceptive, in the sense that they receive physical or chemical signals from the surrounding environment, perceived by capturing them with suitable sensors, feeling them, in the sense that they are processed through biochemical and biophysical pathways, and react appropriately to them [31]. Many animals are conscious, that is they respond to external stimuli, as perceptive beings, through a mental project aimed at optimising actions and reactions designed towards the perpetuation of the species (Darwin docet) [32]. Therefore, livestock animals are undoubtedly conscious beings: in fact, to perform an invasive manipulation on a bull, for example, to surgically extract an annoying cyst, we have to make it unconscious with sedatives or anaesthesia, otherwise it will be the bull who will put us to sleep. Awareness (in particular self-awareness), on the other hand, is a very rare trait in animals and concerns the possibility of recognising oneself as an active agent in relation to the surrounding world. To discover this faculty, zoo-psychologists use the mirror test which has been passed fully only by chimpanzees [33] and some corvids, although the results are still controversial [34] (the domestic dog, for example, does not pass the test, but recognised themselves when the test was enhanced with the aid of smells [35]). Finally, elaboration of mental representations is the ability to deal with symbols and to compare them, distinguishing them from reality (or phenomenic, even if the word is demanding). This ability is unique to our species, *Homo sapiens*, and likely to be extended to some extent to other hominins, but the debate among paleoanthropologists is ongoing. The development of linguistic skills has played a major role in the development of this ability [36].

That being said, what are the necessary conditions for animals to be bearers of rights? If rights (and specular duties [37]) are a human construct based on a social pact—which codifies constitutions, laws, courts, forms of representation, etc.—then only forms of life endowed with the ability to make mental representations can arrive at a highly symbolic outcome such as that constituted by the conventional rules that regulate human communities. Therefore animals, all animals, cannot hold rights as they are not part of the symbolic community made up only of human beings. The absence of laws, courts, judges, and lawyers of animals for animals implies that the notion of animal rights is not applicable and that it is solely our responsibility to ensure their welfare. These obligations extend not only to animals, but also to other living beings, either single or inserted in ecosystems. This asymmetry, the fact that animals are not capable of claiming their rights, like other animate categories present in nature, such as living organisms in ecosystems, or inanimate ones present in nurture (think of the duties of preserving historical and archaeological memories), but which have a value in our moral sphere, implies that these duties are balanced by as many rights that pertain to the domain of humans, even if placed in different planes: the right to protect animals, the right to protect the environment, the right to preserve cultural heritage. The responsibility to provide a life free from unnecessary suffering and to attend to their needs lies with us as human beings. This responsibility is similar to our other obligations such as ensuring clean air, education, and respect for nature. This relationship between duties and rights is what forms the basis for the argument in favour of the permissibility of animal breeding for human purposes.

It does not escape notice that the correlativity between duties and rights is usually in the hands of different subjects, the former generally being the responsibility of states and the latter typically individual. However, states that are organised according to norms recognise in the responsibility of individuals the fulfilment of duties resulting from rights, at least under the modern forms of democratic order. This means that the respect and protection of animals, assumed to be a citizen’s right, must be enforced by the state in the form and manner prescribed by constitutions and laws. Several constitutionalists agree that animals do not have constitutional rights, but an increasing number of constitutions provide for the protection of non-human animals, and Eisen [38] predicts that evermore will do so in the future. A right that has recently been included in the Italian Constitution (Article 9, the Italian Republic recognizes the protection of the environment, biodiversity and ecosystems by regulating the methods and forms of animal protection).

The lack of passive rights for animals is also a consequence of the general lack of responsibility for their actions. According with Musschenga [39] “humans are responsible for their habitual behaviour if they have the capacity for deliberate intervention. Although animals are capable of intervention in their habitual behaviour, they are not capable of deliberate intervention”. This means that, even though animals are intentional agents in that they achieve, in the most evolved forms of mental states, their own precise purposes, if an animal causes harm to a human, the responsibility lies with the owner, in the case of domestic animals, as they are obliged to take care of them, and normally for wild animals it lies with the state, (i.e., the community to which they belong). However, this position is quite recent, so much so that animal courts existed from antiquity to the Middle Ages [40] and some contemporary philosophers hold animals morally responsible for the harm caused to humans [41].

## 6. Is it Permissible to Breed and Sacrifice Animals to Our Own Ends? Yes, But

At the same level as the human right to respect animals and the environment is the right to have access to good and sufficient food, which we share with all species through our instinctual need for survival. This right becomes a primary one when it is recognized as such through the Universal Declaration of the Right to Food [42]. This right gives us the ability to demand a healthy and accessible diet for all people, and animal-derived products are an essential part of fulfilling this fundamental right, which is considered a universal right for all humanity [43].

At this point one could object that our species could survive very well without consuming animal-based foods. This position is immediately refuted by the medical observation for the need of artificial supplements in vegan diets and clearly serves as evidence of the biological requirement for the consumption of meat, milk, eggs, and fish, and this forms the foundation of the right to food [44,45,46]. As long as this is done with total respect for the animals and with the guarantee of their welfare.

It’s also worth noting that the arguments used by supporters of animal “passive rights”, evidencing marginal cases and equating animals with people who cannot communicate or are in a permanent state of unconsciousness, is misleading. This argument, known as the Argument from Marginal Cases (AMC), is a flawed reasoning that can justify any action. This type of thinking is central to animal rights activists who equate the moral status of animals (which ones?) with that of humans without speech or consciousness, but the consequences of such equality are difficult to predict and imagine, either positively or negatively [47].

Therefore, we must raise livestock to respond to one of our primary rights, food [48], without however failing another right of ours, the respect and well-being of animals at all stages of life, which corresponds to our relative duties both towards them, but also towards the environment and society as a whole. Our right not to make livestock animals suffer must be taken, as it is already in many states, as a rule of law that criminalises those who unjustifiably cause suffering to animals, abandon them, overwork them, use them for their own amusement causing suffering. Just as we are obliged, as citizens, to report crimes against humans to authorities, we all must be obliged to report crimes and mistreatments against animals. However, in order to avoid using the marginal reasoning that all animals have the same protection status, we must be clear on which animals and in what situations this collective obligation must be exercised. For example, protecting laboratory rats and exterminating sewer rats with poison, in the first case entails strict rules of behaviour for scientists, in the second case, the expenditure of public money for municipalities: but these are the same species. Or again, the protection of bees is considered an obligation, while the expulsion of wasps from homes a necessity. Yet again, and the examples could be endless, turning a hectare of pasture where two steers were bred and slaughtered into a wheat crop involving the sacrifice of more than 40 small animals at the same level of cognitive evolution [49]. Finally, to remain in the field of food production, the fight against insects and other predators of our food is a practice largely justified by the need to safeguard the primary good represented by human nutrition and healthy food [50].

Having clarified that it is legitimate to raise animals for our food requirements, we can do it in an ethical and sustainable way, with techniques and methods capable of safeguarding the welfare of animals, defending them primarily against pain and disease, taking into account that livestock represents an economic value of strategic importance in developed countries and the only way to survive for hundreds of millions of people in developing countries [51].

## 7. Conclusions

To the question of whether it is acceptable to breed and sacrifice animals to produce food (but also for companion animals, laboratory animals, sports animals, etc.) my reasoning leads to a positive answer, as this responds to our primary rights or enters into the sphere of respect for the animals we have domesticated and selected as our right, albeit not a vital one. The necessary condition is that these rights of ours are not detrimental to our other equally important right, that of respecting animals. This right is not only a moral but also a legal obligation, and its respect is expressed in the ethical and regulatory dictate that no harm be done to the animals for which we are responsible.

This new right, ontologically different from the passive rights of animals that many claim are real and verifiable, must be upheld wherever possible, even sacrificing some of the zeal with which it is often upheld.

Finally, the central theme that has animated this reflection remains our profound responsibility towards all living beings and ecosystems more generally, the care and respect of which must be considered for us as absolute values of humanity.

## Data Availability

Not applicable.

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
