# Peer review of "Anthropocentrism, Natural Harmony, Sentience and Animal Rights: Are We Allowed to Use Animals for Our Own Purposes?"

_animals, 2023, doi:10.3390/ani13061083_

Round 1

Reviewer 1 Report

In this opinion piece, Giuseppe Pulina advocates that the only rational and legitimate view on our obligations towards animals is an anthropocentric one. He further challenges a previous paper by James Yates, covering the concepts of harmony with nature and recognition of its values.  

I do not think I should weigh in on whether an opinion should or not be published. However, I do urge the author to reflect on the manuscript as it currently stands, acknowledge its many shortcomings, and revise it accordingly, if he so wishes.

Before expressing the shortcomings I have found in this manuscript, it is only fair that I acknowledge where I agree with the author. Firstly, I agree that such loosely-defined concepts as “harmony” should be replaced by more concrete ones, unless they are properly defined or disambiguated from other meanings for a given field or discussion.

Second, it is undeniable that, as far as we know, only humans feel moral obligations towards animals, which arise from our greater understanding of the ethical consequences of our actions. We are, of course, quite contradictory in our assessment of duties and responsibilities in face of the suffering of sentient creatures (some animals we love, some we hate, some we eat, as Hal Herzog pointed out). However, this does not mean that whatever stance humans have in regard to animal ethics, due to its anthropogenicity, must be necessarily anthropocentric.

 In regard to its shortcomings, I will address them by order of appearance in the paper.

Ln 20 – How come “sentience” is presented as “an ambiguous term”? If any being can subjectively experience feelings, such as pain, pleasure and anything in between, then said being is sentient, by most accounts.

Ln 21 - The claim that “there can be only one perspective, the anthropocentric one” is ascertained without any substantial grounding, and should therefore equally be dismissed. I should be added, however, that most animal ethics schools of thought are not anthropocentric, which in and of itself is consequence of this fact.

Ln 24 – It seems lost to the author that if humans have ‘an obligation’ to properly care for and protect animals, then we are entitling them the “right” to said protection, regardless of whether it stems from an intrinsic right of animals, or one that is bestowed upon them. Also, if something is an obligation, it is not a right.

Ln 29-33 – This sentence goes nowhere.

Ln 47 – Why is ‘anthropocentrism’ wrongly used to describe a human-centred viewpoint? It is quite literally what anthropocentric means. Antrophos (human) + centrum (centre).

Ln 65-71 – While one should be cautious to not project exclusively human experiences towards other species, this is a far cry from believing that all mental states humans experience are exclusive to our species. This would be against our shared ancestry, anatomy, and physiology, and lest we forget we are animals too.

Ln 70 – What is a “semantic rave”?

Ln 73 – Why assume that the terms we use to describe human emotions and experiences will be replaced for “more correct terms” for non-humans based on the advancement of cognitive science? It might actually go the other way around, and indeed we end up absolutely confirming what many already strongly believe, that other species can experience not only basic emotions such as pain, fear or desire, but also more complex feelings like boredom, joy, depression or empathy.

Ln 80 – While James Yeates refers to the “values of Nature”, there is nothing in his text that suggests that he attributes any level of sentience of conscience of said values, but rather that it has value beyond the one us humans attribute to it. If I’m not mistaken, the point of the author is that if there were no humans, there would be no one to value nature, and therefore, only from an anthropocentric point of view can nature have value. While I tend to shy away from any concept of ‘intrinsic’ value, or natural right, even of humans, I think one should not, however, exclude the role of animals as “valuers” of nature. If us humans were not here, other species would, which would value nature. And excluding animals from being able to value anything, in any capacity, would be hard to justify.

Lines 100-103 – I would expect more from the author than to resort to a slippery slope fallacy.

Line 105 – Not sure if “prejudicially established” helps in getting the authors’ point of view across.

Line 112-114 – While I agree wholeheartedly that we do have moral obligations towards animals, including of minimizing suffering as much as possible, the author claims that this should be done from an anthropocentric point of view, which I can only interpret to be an action for the sake of humans. However, one should treat animals with respect for the sake of animals themselves.

Ln 118 – Please consider revising this sentence. Also, I believe the author means “preclinical”, rather than clinical trials. Moreover, preclinical trials are but a fraction of animal use for biomedical purposes.

Ln 127 – Please consider replacing “homo-animal” for “human-animal”. “homo” is a prefix that means “the same” (homology, homogenous, and so on), rather than human.

Ln 144 – Please consider adding “of” between “question” and “whether”

Ln 160 – The author is probably not acquainted with “feral” animals, i.e. domestic animals behaving in such a way that is akin of their closest wild counterparts. Indeed, I recommend the documentary “The Laboratory Rat: A Natural History”, by Manuel Berdoy, which clearly shows that while we have taken animals from the wild, we are far from taking the wild from the animals.

Ln 165-166 – It would be more useful to discuss what we mean by “rights”, in this context. As I said before, if one grants animals protection, one can argue we are granting them “a right to be protected”, even if it these are “limited rights”. For instance, by establishing an upper threshold of suffering beyond which we are not to allow animals to continue in an experiment, we are granting an animal a right, of sorts, (albeit not as Tom Regan has postulated them) to not suffer more than said limit. So, the whole answer to the question “do animals have rights?” can be boiled down to whether humans grant such rights or not, if we accept that rights are claimed or granted, but not intrinsic to any being or population.

Ln 176-178 – Quite ironically, the author falls into the trap of anthropomorphism, by only acknowledging conscience if it fits into our concept of it.

Ln 184 – Is the author aware of any extant Homo species, other than Homo sapiens? Also, I suggesting putting “Homo” in italics.

Ln 193 – Why are animals excluded from having mental representations? Many animal species have memories, plan ahead, learn, even negotiate with each other.

Ln 195-198 – This is just ungrounded, by all measures. Even if only us humans are indeed capable of claiming and/or granting rights, it does not exclude us to expand the circle of moral consideration to give other animal species rights. Even if neither all species nor all rights, and even if inconsistently (we are, quite unfortunately, equally inconsistent in our acknowledgement of human rights). Moreover, long before there were laws, courts, lawyers, or scholarship, there were already humans with the same mental capabilities we have nowadays. So there was a time when the concept of rights had not even been “invented”, yet humans had all the traits necessary for benefitting or acknowledge someone else's rights. And since rights are an invention anyway, rather than an intrinsic property of humans, we can invent them in such a way to include other species.

Ln 200 – Why does the author state that “the existence of duties towards animals stems from the absence of their ability to have rights“? Such a claim, while bold, is not grounded on anything of substance.

Ln 222 – Why deny animals agency? Makes no sense.

Ln 227-228 – Again, the slippery slope fallacy. Not particularly sophisticated.

Ln 237-241 – Even if it were entirely true that a healthy vegan diet is not possible, one can benefit from animal protein without sacrificing animals, or even jeopardizing their wellbeing, if animal-based products are obtained from animals under good welfare conditions.

Ln 253 – Not making livestock animals suffer is not a human right, but rather a human obligation.

Ln 271-276 – While I agree that it is possible to breed animals in “an ethic and sustainable way, with techniques and methods capable of safeguarding the welfare of animals, defending them primarily against pain and disease, preserving the environment, respecting cultures, health and landscape”, I missed where in the text the author was successful in establishing this, as a matter of fact. Indeed, the text as a whole is not structured and developed in such a way that it leads to the manuscript’s conclusion.

The author should also mind that, somewhere in the text, the quotations get misaligned with the references. 

Author Response

Reviewer 1

In this opinion piece, Giuseppe Pulina advocates that the only rational and legitimate view on our obligations towards animals is an anthropocentric one. He further challenges a previous paper by James Yates, covering the concepts of harmony with nature and recognition of its values. 

I do not think I should weigh in on whether an opinion should or not be published. However, I do urge the author to reflect on the manuscript as it currently stands, acknowledge its many shortcomings, and revise it accordingly, if he so wishes.

Before expressing the shortcomings I have found in this manuscript, it is only fair that I acknowledge where I agree with the author. Firstly, I agree that such loosely-defined concepts as “harmony” should be replaced by more concrete ones, unless they are properly defined or disambiguated from other meanings for a given field or discussion.

Second, it is undeniable that, as far as we know, only humans feel moral obligations towards animals, which arise from our greater understanding of the ethical consequences of our actions. We are, of course, quite contradictory in our assessment of duties and responsibilities in face of the suffering of sentient creatures (some animals we love, some we hate, some we eat, as Hal Herzog pointed out). However, this does not mean that whatever stance humans have in regard to animal ethics, due to its anthropogenicity, must be necessarily anthropocentric.

 In regard to its shortcomings, I will address them by order of appearance in the paper.

  1. I thank the Referee for having meticulously read my work and having faced the difficult task of judging an opinion paper trying not to make the mistake of putting his opinions before those expressed by me. I have carefully considered the objections and suggestions that the Referee has provided and I have agreed with many while answering a few others, I have not agreed, as clearly as possible for me. However, all of them have contributed to enormously improve the paper and to make it more suitable for possible publication in a scientific journal such as animals. On the anthropomorphism-anthropocentrism theme, I think I did not express my thought clearly, and I thank the Referee for this: I tried in the text to better clarify the consequentiality between the two terms, also correcting the ambiguity that could lead us to believe that two terms are synonyms.

Ln 20 – How come “sentience” is presented as “an ambiguous term”? If any being can subjectively experience feelings, such as pain, pleasure and anything in between, then said being is sentient, by most accounts.

  1. Accepted. The word was substituted

Ln 21 - The claim that “there can be only one perspective, the anthropocentric one” is ascertained without any substantial grounding, and should therefore equally be dismissed. I should be added, however, that most animal ethics schools of thought are not anthropocentric, which in and of itself is consequence of this fact.

  1. Accepted. The phrase was rewritten

Ln 24 – It seems lost to the author that if humans have ‘an obligation’ to properly care for and protect animals, then we are entitling them the “right” to said protection, regardless of whether it stems from an intrinsic right of animals, or one that is bestowed upon them. Also, if something is an obligation, it is not a right.

  1. On this point I do not agree totally with the Referee. The summary briefly describes the content of the paper and a large part of the discussion hinges on the point of animal rights. Therefore this part of the summary I cannot change, as well as, with arguments and better exposition, I will keep the point on the lack of animal rights.

Ln 29-33 – This sentence goes nowhere.

  1. Accepted.

Ln 47 – Why is ‘anthropocentrism’ wrongly used to describe a human-centred viewpoint? It is quite literally what anthropocentric means. Antrophos (human) + centrum (centre).

  1. Accepted. The phrase was rewritten

Ln 65-71 – While one should be cautious to not project exclusively human experiences towards other species, this is a far cry from believing that all mental states humans experience are exclusive to our species. This would be against our shared ancestry, anatomy, and physiology, and lest we forget we are animals too.

  1. Accepted. The phrase was rewritten

Ln 70 – What is a “semantic rave”?

  1. Mistake. Thank you

Ln 73 – Why assume that the terms we use to describe human emotions and experiences will be replaced for “more correct terms” for non-humans based on the advancement of cognitive science? It might actually go the other way around, and indeed we end up absolutely confirming what many already strongly believe, that other species can experience not only basic emotions such as pain, fear or desire, but also more complex feelings like boredom, joy, depression or empathy.

  1. I do not agree with the Referee. The emotional spectrum is measurable with extreme difficulty even in humans, as demonstrated by this founding and much cited paper (https://www.ncbi.nlm.nih.gov/pmc/articles/PMC2756702/pdf/nihms134765.pdf) in which in the first reads the following sentence: “scientific evidence suggests that measuring a person’s emotional state is one of the most vexing problems in affective science”.

I believe that the word "sentience" as an aggregate of feelings is difficult to use in the human sphere and even more so in the animal one. However, if we continue to use abstract concepts of the emotional sphere that are difficult to measure, we will bring a qualitative contribution, albeit important to science, but we will not face the problem of quantifying these mental states.

Ln 80 – While James Yeates refers to the “values of Nature”, there is nothing in his text that suggests that he attributes any level of sentience of conscience of said values, but rather that it has value beyond the one us humans attribute to it. If I’m not mistaken, the point of the author is that if there were no humans, there would be no one to value nature, and therefore, only from an anthropocentric point of view can nature have value. While I tend to shy away from any concept of ‘intrinsic’ value, or natural right, even of humans, I think one should not, however, exclude the role of animals as “valuers” of nature. If us humans were not here, other species would, which would value nature. And excluding animals from being able to value anything, in any capacity, would be hard to justify.

  1. Accepted.A sentence has been added to better explain the meaning of nature values

Lines 100-103 – I would expect more from the author than to resort to a slippery slope fallacy.

  1. The sentence judged slippery slope fallacy is the following: “Unless one wants to espouse the position of Mikhalevich, and Powell [9] where all animals, vertebrates and invertebrates, should be included in the category of beings to be protected, a kind of absurd pan-animalism whereby even parasites and disease vectors should be safeguarded.”. Without going into the merits of the complexity of the definition, as reported by [Douglas Walton. Informal Logic, Vol. 35, No. 3 (2015), pp. 273 –311], I limit myself to a definition considered acceptable by this author “Volokh (2002, 1030): “I think the most useful definition of a slippery slope is one that covers all situations where decision A, which you might find appealing, ends up materially increasing the probability that others will bring about decision B, which you oppose.” The fact is that I do not consider the position of Mikhalevich, and Powell neither attractive nor justified [a kind of absurd pan-animalism], and I reinforce my position by pointing out the absurd consequences of considering 'all animals' to be subjects of rights. In a logical sense, I assert that A is wrong when applied to B so I conclude (C) that position A is absurd.

However, I accept the suggestion to make the thought more explicit, and have rewritten the sentence more clearly.

Line 105 – Not sure if “prejudicially established” helps in getting the authors’ point of view across.

  1. Accepted. Good point, thank you. The phrase was changed accordingly

Line 112-114 – While I agree wholeheartedly that we do have moral obligations towards animals, including of minimizing suffering as much as possible, the author claims that this should be done from an anthropocentric point of view, which I can only interpret to be an action for the sake of humans. However, one should treat animals with respect for the sake of animals themselves.

  1. Accepted. I have enclosed this concept in the phrase.

Ln 118 – Please consider revising this sentence. Also, I believe the author means “preclinical”, rather than clinical trials. Moreover, preclinical trials are but a fraction of animal use for biomedical purposes.

  1. Accepted, correct! I expanded the meaning of the use of animals in pharmaceutical and medical experimentation

Ln 127 – Please consider replacing “homo-animal” for “human-animal”. “homo” is a prefix that means “the same” (homology, homogenous, and so on), rather than human.

  1. Mistake. Thank you.

Ln 144 – Please consider adding “of” between “question” and “whether”

  1. Thank you

Ln 160 – The author is probably not acquainted with “feral” animals, i.e. domestic animals behaving in such a way that is akin of their closest wild counterparts. Indeed, I recommend the documentary “The Laboratory Rat: A Natural History”, by Manuel Berdoy, which clearly shows that while we have taken animals from the wild, we are far from taking the wild from the animals.

  1. In my career, I have managed vast nature park areas and flora and fauna reserves (more than 250,000 ha). I also managed depopulation of both wild (mainly wild boar) and domestic (rewild cattle) fauna for many years. I also ran wildlife rehabilitation hospitals, with all the ethical problems associated with training predators to hunt to release them back into the wild. What I have stated is that domestic animals have been selected for behavioural traits different from those they express in nature, so much so that the breed standards for dogs also include a reference to behaviour considered normal for each breed.

However, I have added a clarification to the sentence to better specify the behavioural retrogression of domestica animals abandoned

Ln 165-166 – It would be more useful to discuss what we mean by “rights”, in this context. As I said before, if one grants animals protection, one can argue we are granting them “a right to be protected”, even if it these are “limited rights”. For instance, by establishing an upper threshold of suffering beyond which we are not to allow animals to continue in an experiment, we are granting an animal a right, of sorts, (albeit not as Tom Regan has postulated them) to not suffer more than said limit. So, the whole answer to the question “do animals have rights?” can be boiled down to whether humans grant such rights or not, if we accept that rights are claimed or granted, but not intrinsic to any being or population.

  1. Accepted. I have introduced the concept of rights as used in the paper.

Ln 176-178 – Quite ironically, the author falls into the trap of anthropomorphism, by only acknowledging conscience if it fits into our concept of it.

  1. Thank you for your sophisticated observation. Actually, awareness is a state well determined by cognitive science and measurable with the aforementioned mirror test. All the paradigmatic premises and methods of study are well described in the specialist literature. I have taken note of them, verifying only that the possession of this faculty in animals can only be done in comparison with humans. And this reasoning is part of the path I have taken to demonstrate the differences rather than the similarities between humans and animals. For the latter, there is a vast literature, which I and others attempt to refute, that places them precisely as the basis for the possession of rights by animals.

Ln 184 – Is the author aware of any extant Homo species, other than Homo sapiens? Also, I suggesting putting “Homo” in italics.

  1. Thank you.

Ln 193 – Why are animals excluded from having mental representations? Many animal species have memories, plan ahead, learn, even negotiate with each other.

  1. Thank you. In fact, the concept was badly expressed and I have clarified it. It is about processing mental representations in a symbolic manner

Ln 195-198 – This is just ungrounded, by all measures. Even if only us humans are indeed capable of claiming and/or granting rights, it does not exclude us to expand the circle of moral consideration to give other animal species rights. Even if neither all species nor all rights, and even if inconsistently (we are, quite unfortunately, equally inconsistent in our acknowledgement of human rights). Moreover, long before there were laws, courts, lawyers, or scholarship, there were already humans with the same mental capabilities we have nowadays. So there was a time when the concept of rights had not even been “invented”, yet humans had all the traits necessary for benefitting or acknowledge someone else's rights. And since rights are an invention anyway, rather than an intrinsic property of humans, we can invent them in such a way to include other species.

  1. I am grateful for the Referee's point of view, but this is the core of my argument that I am willing to discuss at length. In preclassical times, the concept of rights was only a prerogative of the noble and, to some extent, military classes. With the Greeks and Romans, rights were applied to things and persons, taking care that animals were considered both on a par with slaves. Therefore, in classical times, animals were subject (from subjectum, subjected) to law, but were equally held responsible and judged by courts. The barbarian laws and medieval codes downgraded animals to res, an as a things entrusted their responsibility entirely to their owners. This approach has transcended into the modern and contemporary conception of law, for which there is no liability and animals are subject to our protection in different ways in different legal systems. In some of which traces of the subjective responsibility of animals remain, and in others new trends are emerging such as the habeas corpus of a recent judgement granted to a chimpanzee (https://www.nonhumanrights.org/blog/judge-recognizes-two-chimpanzees-as-legal-persons-grants-them-writ-of-habeas-corpus/).

I agree that rights are a rather recent invention of ours, and one that underpins the civilisation of historical societies compared to pre- or proto-historical ones; but for this reason, they must be handled with care and, if granted, by those who can guarantee their respect, after careful and long reflection.

Ln 200 – Why does the author state that “the existence of duties towards animals stems from the absence of their ability to have rights“? Such a claim, while bold, is not grounded on anything of substance.

  1. Accepted, thank you. The sentence misspoke the concept that the fact that animals are not capable of claiming their rights, like other animate categories present in nature, such as living organisms in ecosystems, or inanimate ones present in nurture (think of the duties of preserving historical and archaeological memories), but which have a value in our moral sphere, implies that these duties are balanced by as many rights that pertain to the sphere of humans: the right to protect animals, the right to protect the environment, the right to preserve cultural heritage. Different rights and placed on different planes, but always referable to our capacity to be agents in fulfilling them and sanctioning those who do not respect them.

Ln 222 – Why deny animals agency? Makes no sense.

  1. I do not deny the possibility of action for animals, but I discuss the consequences in terms of liability if the actions of animal’s cause harm to humans. However, I accept the objection and improve the comprehensibility of the sentence, thank you

Ln 227-228 – Again, the slippery slope fallacy. Not particularly sophisticated.

  1. In this case the Referee is right: I have made the sentence explicit so that it cannot be confused with a fallacy. Thank you

Ln 237-241 – Even if it were entirely true that a healthy vegan diet is not possible, one can benefit from animal protein without sacrificing animals, or even jeopardizing their wellbeing, if animal-based products are obtained from animals under good welfare conditions.

  1. If I understand the objection correctly, it is possible to obtain animal protein without sacrificing animals or mistreating them. I have added a sentence clarifying this point according to the remark, thank you.

Ln 253 – Not making livestock animals suffer is not a human right, but rather a human obligation.

  1. Disagreeing on this point is the consequence of what was argued before. Not causing suffering in animals is a moral category of ours (animals do not possess it and in fact nature is a place of suffering!) from which derives the value and consequent right not to see animals mistreated (the right of all citizens). Those who have a duty not to mistreat animals are all of us, but those who have a duty to prosecute perpetrators are the rule of law.

Ln 271-276 – While I agree that it is possible to breed animals in “an ethic and sustainable way, with techniques and methods capable of safeguarding the welfare of animals, defending them primarily against pain and disease, preserving the environment, respecting cultures, health and landscape”, I missed where in the text the author was successful in establishing this, as a matter of fact. Indeed, the text as a whole is not structured and developed in such a way that it leads to the manuscript’s conclusion.

  1. Accepted, I have deleted part of the phrase.

The author should also mind that, somewhere in the text, the quotations get misaligned with the references.

  1. Thanks for the tip. In fact, the automatic format layout had misaligned some references. I checked and corrected them

Reviewer 2 Report

This work is not a classic Article. It is a reflection on an ethical question. On this subject all positions can be legitimate if properly justified and yet remain the expression of a point of view.

The work needs to be reviewed trying to broaden one's point of view, while not denying one's thesis.

Phrases such as "there is no doubt that it is clear that humans are the only beings with the ability to express a point of view due to their possession of language and consequently have the tools to express a point of view"(line 56-58), need to be rethought since language is not only a human possession. It is amply demonstrated that animals also have a language. The fact that man is still unable to understand the language of the various animal species does not make man the only species capable of expressing an opinion.

Therefore I believe that the article can be improved by a critical re-reading that does not fall into the error of attributing to human more rights than the natural rules allow him.

The conclusions probably deserve a broader reflection and a less net indication

Author Response

Rewier 2

This work is not a classic Article. It is a reflection on an ethical question. On this subject all positions can be legitimate if properly justified and yet remain the expression of a point of view.

The work needs to be reviewed trying to broaden one's point of view, while not denying one's thesis.

Phrases such as "there is no doubt that it is clear that humans are the only beings with the ability to express a point of view due to their possession of language and consequently have the tools to express a point of view"(line 56-58), need to be rethought since language is not only a human possession. It is amply demonstrated that animals also have a language. The fact that man is still unable to understand the language of the various animal species does not make man the only species capable of expressing an opinion.

Therefore I believe that the article can be improved by a critical re-reading that does not fall into the error of attributing to human more rights than the natural rules allow him.

The conclusions probably deserve a broader reflection and a less net indication

  1. I thank the Referee for reading the work and providing suggestions on how to improve it. I agree with him that an opinion paper is very difficult to judge by the classical scientific standards without falling into the fallacy of expressing one's own opinion over those of the author. I welcome all the suggestions made and reconsider several parts of the text to make it more comprehensible and better justify the observations. Regarding the pointed remark on language, I agree that other species also have linguistic codes, sometimes very refined ones; however, I am not aware that these are organised in the word as we understand it when we speak of human language. I have also prefixed this point so as not to create ambiguity. Finally, I have taken up some concepts in the conclusions to make them less assertive and more open to different points of view.

Reviewer 3 Report

It is a pity that despite the potential limitations of the article mentioned by the author, a more thorough analysis of the problem has not been made. Nevertheless, I believe that even in its current form, the manuscript is a valuable source of polemics between the supporters of the animal rights theory and the typical welfare approach to the human-animal relationship.

Author Response

Thank you for your comment, which I welcome. It is true that I could have commented and countered the quoted article in more depth, but my purpose was to take my cue from it to try to introduce the topic of the legitimacy of granting rights to animals. I have also tried to be brief enough to respect the style of the journal and not bore any readers too much. I hope that the debate that will be provoked will lead to a better definition of all the points touched on in it. However, I attach the R1 file with some parts better detailed and explained. 

Round 2

Reviewer 1 Report

I provided some comments on the previously submitted version of this manuscript in regard to the clarity of the reasoning and the grounding of some assumptions and conclusions. Some of these comments were duly addressed, improving the manuscript, while others were not, with the author providing arguments as to why this was so. Hence, and while I continue to have divergences of opinion regarding a substantial part of what is written in this manuscript, these are not in any way an impediment to its publication. As I had stated in my previous review, I do not believe any reviewer should gatekeep any paper from being published only because they forward opinions divergent from their own (unless, of course, they were against fundamental human principles, which is of course not the case, here). Doing so would infringe on the most fundamental principles of academic integrity. 

I have only the following minor suggestions:

- mind the typo in line 117

- consider replacing "union" for "relationship", "interaction" or something of the sort, in line 142

- If you are to refer to animalism as an "ideology-religion" (line 143), it is best to back it up with a reference. Perhaps Jacobsson 2014 (DOI:10.3384/cu.2000.1525.146305), Jamison et al 2000 (10.1163/156853000511140), or Friedrich 2014 (https://pil.law.harvard.edu/wp-content/uploads/2016/04/Church-of-Animal-Liberation-Friedrich.pdf). It should be noted, however, that not all animalists hold beliefs or express and act on them as if they were members of a kind of secular religion (indeed I was risk saying most do not).

Author Response

I provided some comments on the previously submitted version of this manuscript in regard to the clarity of the reasoning and the grounding of some assumptions and conclusions. Some of these comments were duly addressed, improving the manuscript, while others were not, with the author providing arguments as to why this was so. Hence, and while I continue to have divergences of opinion regarding a substantial part of what is written in this manuscript, these are not in any way an impediment to its publication. As I had stated in my previous review, I do not believe any reviewer should gatekeep any paper from being published only because they forward opinions divergent from their own (unless, of course, they were against fundamental human principles, which is of course not the case, here). Doing so would infringe on the most fundamental principles of academic integrity.

  1. Thank you very much. I have added to the thanks to the referees the sentence specifying that the opinions expressed in the paper are mine alone.

I have only the following minor suggestions:

- mind the typo in line 117

  1. Corrected, thank you.

- consider replacing "union" for "relationship", "interaction" or something of the sort, in line 142

  1. Replaced, thank you

- If you are to refer to animalism as an "ideology-religion" (line 143), it is best to back it up with a reference. Perhaps Jacobsson 2014 (DOI:10.3384/cu.2000.1525.146305), Jamison et al 2000 (10.1163/156853000511140), or Friedrich 2014 (https://pil.law.harvard.edu/wp-content/uploads/2016/04/Church-of-Animal-Liberation-Friedrich.pdf). It should be noted, however, that not all animalists hold beliefs or express and act on them as if they were members of a kind of secular religion (indeed I was risk saying most do not).

  1. Thank you. I added the suggested reference

Reviewer 2 Report

The author welcomed all suggestions and made the requested changes. The article can be published

Author Response

The author welcomed all suggestions and made the requested changes. The article can be published

  1. Thanks again for the suggestions that improved the article.